# Asymmetric Henry Reaction of 2-Acylpyridine *N*-Oxides Catalyzed by a Ni-Aminophenol Sulfonamide Complex: An Unexpected Mononuclear Catalyst

**DOI:** 10.3390/molecules24081471

**Published:** 2019-04-14

**Authors:** Mouxiong Liu, Dongdong Gui, Ping Deng, Hui Zhou

**Affiliations:** School of Pharmaceutical Science, Chongqing Medical University, Chongqing 400016, China; mouxiongliu@163.com (M.L.); g420646114@163.com (D.G.); 100865@cqmu.edu.cn (P.D.)

**Keywords:** asymmetric catalysis, Henry reaction, ketones, *N*-oxides, aminophenol sulfonamide

## Abstract

The asymmetric Henry reaction of 2-acylpyridine *N*-oxide remains a challenge as *N*-oxides generally act as competitive catalyst inhibitors or displace activating ligands. A novel variable yield (up to 99%) asymmetric Henry reaction of 2-acypyridine *N*-oxides catalyzed by a Ni-aminophenol sulfonamide complex with good to excellent enantioselectivity (up to 99%) has been developed. Mechanistic studies suggest that the unique properties of the electron-pairs of *N*-oxides for complexation with Ni makes the unexpected mononuclear complex, rather than the previously reported dinuclear complex, the active species.

## 1. Introduction

The asymmetric construction of chiral quaternary stereocenters represents a considerable challenge in modern organic synthesis [1,2,3,4,5,6]. The Henry (nitroaldol) reaction [7,8,9,10,11,12,13] of ketones has become one of the most important and versatile reactions for the construction of quaternary carbons containing hydroxyl and nitro groups. In recent years, considerable effort has been devoted to the development of efficient chiral catalysts for asymmetric Henry reactions of reactive ketones, such as trifluoromethyl ketones (for selected examples, see ref [14,15,16,17,18]), α-keto esters (for selected examples, see [19,20,21,22,23,24]), α-keto amides (for selected examples, see [25,26]), α-keto-phosphonates [27,28], and glyoxal hydrates [29]. Although Matsunaga and Shibasaki reported the kinetic resolution of racemic derivatives [30], the asymmetric catalytic version of simple ketones has experienced little progress. At the same time, interest in pyridine derivatives has increased dramatically with the discovery of many bioactive compounds [31,32,33] and ligands containing pyridine rings [34,35,36,37,38,39]. Pedro and Blay first extended the Henry reactions to 2-acylpyridine *N*-oxides, which provided a convenient way for synthesizing β-amino *tert*-alcohols substructure bearing a quaternary stereocenter bonded to a 2-pyridyl moiety [40]. The asymmetric Henry reaction of 2-acylpyridine *N*-oxide remains a challenge as *N*-oxides generally act as competitive catalyst inhibitors or displace activating ligands (For examples of related asymmetric Henry reaction using *N*-oxides as ligands, see ref [22,41,42,43]). We recently reported an asymmetric Henry reaction of 2-acylpyridine *N*-oxides catalyzed by a pre-prepared Ni-PyBisulidine complex, and the corresponding results are not satisfactory [23]. Herein, we describe a Ni-aminophenol sulfonamide complex for the asymmetric Henry reaction of 2-acylpyridine *N*-oxides.

## 2. Results and Discussion

### 2.1. Catalytic Asymmetric Henry Reaction

In the preliminary study, the complexes prepared in situ from **L1** (Figure 1) and different metal salts in a 1/2 molar ratio (for examples in asymmetric bimetallic catalysis based on aminophenol sulfonamide ligands, see [44,45,46]) were used to catalyze the asymmetric Henry reaction of 2-acylpyridine *N*-oxide and nitromethane, and Ni(OAc)_2_ gave the best results (see the Appendix A for details). However, in the subsequent molar ratio investigation of metal/ligand, it was found that 1/1 gave better enantioselectivity than 2/1 (Table 1, entry 1 vs. entry 3).

The ratio was investigated intensively with the results summarized in Table 1 (entries 1–7). It was found that increased metal ratio could increase the reactivity (Table 1, entries 1 and 2 vs. 3–7) and excess ligands provided higher *ee* (Table 1, entries 4–7 vs. 1–3). The best ratio of metal/ligand was 1/1.1 with 86% yield and 85% *ee*. We speculate that the excess metal could increase the amount of Ni/*N*-oxide complexes and Ni_2_/**L1** complexes [45], both of which are higher active species with lower selectivity. At the ratio of 1/1.1, a 1/1 coordination complex of Ni/**L1** could be generated to the greatest extent. After the screening of benzenesulfonyl moiety, **L2** was found to be the most promising ligand (Table 1, entries 8–14; Figure 1). The corresponding results of **L9**-**L11** (Table 1, entries 15–17; Figure 1) showed both of the phenolic hydroxyl group and sulphonamide group played a key role in achieving high *ee*.

Next, different bases were examined, with the results shown in Table 2. The tertiary and secondary amines investigated showed excellent activity, except for *N*-methylmorpholine. The substituent size at the nitrogen atoms plays a key role in the selectivity and *N,N*-dicyclohexyl-methylamine gave the best results (Table 2, entry 2). On the other hand, the addition order of 2-acylpyridine *N*-oxide and nitromethane had an effect on the enantioselectivity and the addition of 2-acylpyridine *N*-oxide first was conducive to high *ee* (Table 2, entries 2 vs. 8).

With the optimized reaction conditions in hand (for more detailed results of optimization studies, such as solvents effect, substrate concentration and the amount of nitromethane, see the Appendix A), the substrate scope was explored. The results are summarized in Table 3. The presence of 4- and 5-substituents (Me or Cl) on the pyridine ring did not affect the high activity and excellent selectivity (Table 3, entries 2, 3 and 5). The substituent of 5-Br provided an unexpectedly low yield with a good *ee* (Table 3, entry 6). The 6-position substituent on the pyridine ring greatly impairs the *ee* (Table 3, entry 4). The reaction between 3-methyl substituted substrate and CH_3_NO_2_ did not take place. This catalytic system is still effective for ethyl and propyl ketones (Table 3, entrys 7 and 8). The aromatic ketone afforded product **2i** in good *ee*, albeit with moderate yield (Table 3, entry 9).

### 2.2. Mechanistic Studies of Ni-Aminophenol Sulfonamide Complex

The control experiments (Table 1, entries 1–7) indicated that the mononuclear system is important for high stereoselectivity and the addition of 2-acylpyridine *N*-oxide first was conducive to high *ee* (Table 2, 2 and 8). To gain some insight into the mechanism, the ESI-MS studies of the mixture of Ni(OAc)_2_/**L2** (1:1.1) and **1a** were carried out (Figure 2, for more details, see Appendix A). The spectrum displayed ions at *m*/*z* 1179, 1316, 1453, 1590, which corresponded to **C1**-**C4** (Figure 3). This confirms the unique properties of the electron-pairs of *N*-oxides for complexation with Lewis acids [47,48,49]. In addition, there was a linear relationship between the enantiomeric excess of the Ni(OAc)_2_-**L2** (1:1.1) catalyst and product **2a** (Figure 4). These results suggested that the active species in the present reaction would be a monomeric NiOAc-**L2** catalyst. The proposed working model was illustrated in Figure 5 to rationalize the asymmetric induction. The keto functionality is coordinated to Ni in the more Lewis acidic equatorial position for maximal activation [50,51], whereas the nitronate generated by the base is positioned by the hydrogen bonding.

## 3. Experimental Section

### 3.1. General Information

Commercial reagents were used as purchased. NMR spectra (600 MHz, Bruker, Karlsruhe, Germany) were recorded in the deuterated solvents as stated, using residual non-deuterated solvent signals as the internal standard. High resolution mass spectra were recorded with a Bruker Solari XFT-ICR-MS system. The enantiomeric excess (*ee*) was determined by HPLC analysis (LC-16, Shimadzu, Suzhou, China) using the corresponding commercial chiral column as stated in the experimental procedures at 23 °C with UV detector. Optical rotations were measured on a commercial polarimeter (Autopol I, Rudolph, Hackettstown, NJ, USA) and are reported as follows: [α]_D_^T^ (c = g/100 mL, solvent). The absolute configuration of **2a**–**2d**, **2f**, **2g** and **2i** were assigned by comparison with the sign of optical rotation value found in the literature. The absolute configuration of **2e** and **2h** was determined by analogy.

### 3.2. General Procedure for Catalytic Asymmetric Reaction

The mixture of Ni(OAc)_2_·4H_2_O (0.02 mmol, 10 mol%) and **L2** (0.022 mmol, 11 mol%) was stirred in THF (0.5 mL) at 35 °C for 1 h. Then 2-acylpyridine *N*-oxide (0.2 mmol) and the base (0.04 mmol, 20 mol%) were added. The mixture was cooled to 0 °C. After stirring for 10 min at 0 °C, CH_3_NO_2_ (0.2 mL) and THF (0.3 mL) were added. The mixture was further stirred at 0 °C for the time indicated in Table 3. The resulting solution was purified by column chromatography (EtOH/AcOEt or petroleum ether/AcOEt) on silica gel to afford the products.

*1-Methyl-2-nitro-1-(1-oxido-2-pyridinyl) ethanol* (**2a**), brown oil, 99% yield, 94% *ee*; ^1^H-NMR (CDCl_3_) δ 8.26 (d, 1H, *J* = 6.4), 7.45–7.42 (m, 2H), 7.37–7.35 (m, 1H), 5.35 (d, 1H, *J* = 11.1), 4.82 (d, 1H, *J* = 11.2), 1.79 (s, 3H). $[\alpha ]{}_{D}^{20}$ = +57 (*c* 0.9, MeOH) [lit. [40] $[\alpha ]{}_{D}^{20}$ = +48 (*c* 0.9, MeOH) in 86% *ee*]; HPLC (CHIRALPAK AD-H column, Daicel, Osaka, Japan, hexane/2-propanol = 75/25, flow 1.0 mL/min, detection at 254 nm) t_r_ = 8.7 min (major) and t_r_ = 20.7 min (minor).

*1-Methyl-2-nitro-1-(4-methyl-1-oxido-2-pyridinyl) ethanol* (**2b**), brown solid, 99% yield, 99% *ee*; ^1^H-NMR (CDCl_3_) δ 8.26 (s, 1H), 8.16 (d, 1H, *J* = 6.6), 7.20 (s, 1H), 7.17 (d, 1H, *J* = 6.7), 5.47 (d, 1H, *J* = 11.0), 4.73 (d, 1H, *J* = 10.9), 2.42 (s, 3H), 1.81 (s, 3H). $[\alpha ]{}_{D}^{20}$ = +156 (*c* 0.4, MeOH) [lit. [40] $[\alpha ]{}_{D}^{20}$ = +41 (*c* 0.9, MeOH) in 84% *ee*]; HPLC (CHIRALPAK AD-H column, hexane/2-propanol = 80/20, flow 1.0 mL/min, detection at 254 nm) t_r_ = 8.2 min (major) and t_r_ = 32.8 min (minor).

*1-Methyl-2-nitro-1-(5-methyl-1-oxido-2-pyridinyl) ethanol* (**2c**), brown solid, 99% yield, 97% *ee*; ^1^H-NMR (CDCl_3_) δ 8.13 (s, 1H), 8.04 (s, 1H), 7.30–7.29 (m, 1H), 7.26–7.25 (m, 1H), 5.43 (d, 1H, *J* = 10.9), 4.73 (d, 1H, *J* = 10.9), 2.37 (s, 3H), 1.80 (s, 3H). $[\alpha ]{}_{D}^{20}$ = +181 (*c* 0.4, MeOH) in 97% *ee* [ lit. [40] $[\alpha ]{}_{D}^{20}$ = +60 (*c* 0.6, MeOH) in 81% *ee*]; HPLC (CHIRALPAK AD-H column, hexane/2-propanol = 75/25, flow 1.0 mL/min, detection at 254 nm) t_r_ = 12.4 min (major) and t_r_ = 18.3 min (minor).

*1-Methyl-2-nitro-1-(6-methyl-1-oxido-2-pyridinyl) ethanol* (**2d**), brown solid, 99% yield, 17% *ee*; ^1^H-NMR (CDCl_3_) δ 8.30 (s, 1H), 7.37–7.29 (m, 3H), 5.47 (d, 1H, *J* = 10.9), 4.73 (d, 1H, *J* = 11.0), 2.58 (s, 3H), 1.80 (s, 3H). $[\alpha ]{}_{D}^{20}$ = +21 (*c* 0.4, MeOH) in 17% *ee* [ lit. [40] $[\alpha ]{}_{D}^{20}$ = +109 (*c* 0.9, MeOH) in 55% *ee*]; HPLC (CHIRALPAK AD-H column, hexane/2-propanol = 80/20, flow 1.0 mL/min, detection at 254 nm) t_r_ = 7.7 min (major) and t_r_ = 11.1 min (minor).

*1-Methyl-2-nitro-1-(4-chlorine -1-oxido-2-pyridinyl) ethanol* (**2e**), brown solid, 99% yield, 92% *ee*; ^1^H-NMR (CDCl_3_) δ 8.20 (d, 1H, *J* = 6.9), 7.45 (d, 1H, *J* = 2.9), 7.41 (s, 1H), 7.36 (dd, 1H, *J_1_* = 6.9, *J_2_* = 2.8), 5.40 (d, 1H, *J* = 11.5), 4.85 (d, 1H, *J* = 11.5), 1.80 (s, 3H). ^13^C-NMR (150 MHz, CDCl_3_) δ 150.6, 141.1, 134.9, 126.0, 125.4, 80.0, 72.4, 23.0. HRMS (ESI): *m*/*z* Calcd [C_8_H_10_ClN_2_O_4_]^+^ [M + H]^+^: 233.0324 (Cl^35^), 235.0300 (Cl^37^), Found 233.0323, 235.0290. $[\alpha ]{}_{D}^{20}$ = +52 (*c* 0.5, MeOH); HPLC (CHIRALPAK AD-H column, hexane/2-propanol = 75/25, flow 1.0 mL/min, detection at 254 nm) t_r_ = 6.0 min (major) and t_r_ = 14.2 min (minor).

*1-Methyl-2-nitro-1-(5-bromo-1-oxido-2-pyridinyl) ethanol* (**2f**), brown solid, 26% yield, 84% *ee*; ^1^H-NMR (CDCl_3_) δ 8.42 (d, 1H, *J* = 1.9), 7.57 (dd, 1H, *J_1_* = 8.6, *J_2_* = 1.8), 7.32 (d, 1H, *J* = 8.6), 5.39 (d, 1H, *J* = 11.4), 4.80 (d, 1H, *J* = 11.3), 1.79 (s, 3H). $[\alpha ]{}_{D}^{20}$ = +48 (*c* 0.3, MeOH) [lit. [40] $[\alpha ]{}_{D}^{20}$ = +74 (*c* 0.9, MeOH) in 89% *ee*]; HPLC (CHIRALPAK AD-H column, hexane/2-propanol = 80/20, flow 1.0 mL/min, detection at 254 nm) t_r_ = 9.5 min (major) and t_r_ = 10.7 min (minor).

*1-Nitromethyl-1-(1-oxido-2-pyridinyl)propan-1-ol* (**2g**), brown solid, 86% yield, 92% *ee*; ^1^H-NMR (CDCl_3_) δ 8.29 (d, 1H, *J* = 6.4), 7.46–7.44 (m, 2H), 7.38–7.36 (m, 1H), 5.31 (d, 1H, *J* = 11.4), 4.97 (d, 1H, *J* = 11.4), 2.28–2.22 (m, 1H), 2.12–2.05 (m, 1H), 1.09 (t, 3H, *J* = 7.4). $[\alpha ]{}_{D}^{20}$ = +64 (*c* 0.4, MeOH) [lit. [40] $[\alpha ]{}_{D}^{20}$ = +63 (*c* 1.2, MeOH) in 81% *ee*]; HPLC (CHIRALPAK AD-H column, hexane/2-propanol = 80/20, flow 1.0 mL/min, detection at 254 nm) t_r_ = 12.6 min (major) and t_r_ = 31.9 min (minor).

*Nitromethyl-1-(1-oxido-2-pyridinyl) but-1-ol* (**2h**), brown solid, 67% yield, 69% *ee*; ^1^H-NMR (CDCl_3_) δ 8.28 (d, 1H, *J* = 6.5), 7.47–7.43 (m, 2H, J = 12.3), 7.37–7.35 (m, 1H), 5.28 (d, 1H, *J* = 11.5), 5.02 (d, 1H, *J* = 11.4), 2.20–2.15 (m, 1H), 2.03–1.98 (m, 1H), 1.65–1.59 (m, 1H), 1.46–1.41 (m, 1H), 1.0 (t, 3H, *J* = 7.4).^13^C-NMR (150 MHz, CDCl_3_) δ 148.2, 139.7, 126.9, 124.6, 124.5, 78.2, 73.8, 36.2, 15.1, 13.2. HRMS (ESI): *m*/*z* calcd for C_10_H_14_N_2_NaO_4_^+^ [M + Na]^+^: 249.0846, found 249.0840. $[\alpha ]{}_{D}^{20}$ = +67 (*c* 0.3, MeOH); HPLC (CHIRALPAK IA column, hexane/2-propanol = 85/15, flow 0.8 mL/min, detection at 254 nm) t_r_ = 16.2 min (major) and t_r_ = 19.2 min (minor).

*1-(4-Chlorophenyl)-2-nitro-1-(1-oxido-2-pyridinyl)ethanol* (**2i**), brown solid, 48% yield, 79% *ee*; ^1^H-NMR (CDCl_3_) δ 8.22 (d, 1H, *J* = 6.3), 7.55 (dd, 1H, *J_1_* = 8.1, *J_2_* = 1.8), 7.46 (t, 1H, *J* = 7.7), 7.43–7.41 (m, 2H), 7.39–7.36 (m, 3H), 5.44 (d, 1H, *J* = 12.7), 5.12 (d, 1H, *J* = 12.7). $[\alpha ]{}_{D}^{20}$ = +50 (*c* 0.2, MeOH) [lit. [40] $[\alpha ]{}_{D}^{20}$ = +55 (*c* 0.7, MeOH) in 90% *ee*]; HPLC (CHIRALPAK AD-H column, hexane/2-propanol = 80/20, flow 1 mL/min, detection at 254 nm) t_r_ = 13.4 min (major) and t_r_ = 17.9 min (minor).

## 4. Conclusions

We have developed a new mononuclear Ni-aminophenol sulphonamide complex for the asymmetric Henry reaction of 2-acylpyridine *N*-oxides. The simple experimental protocol affords various optically active pyridine-containing β-nitro *tert*-alcohols in variable yield (up to 99%) with good to excellent enantioselectivity (up to 99%). Mechanistic studies suggested that the unique properties of the electron-pairs of *N*-oxides for complexation with Ni makes the unexpected mononuclear complex, rather than the previously reported dinuclear complex, the active species.

## Figures and Tables

**Figure 1 molecules-24-01471-f001:**
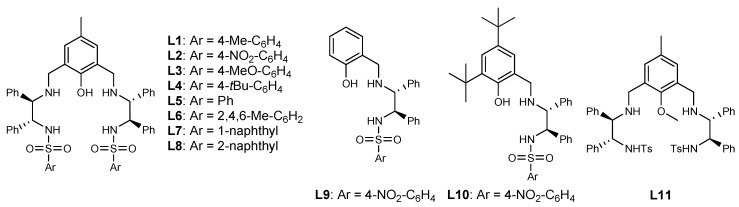
Structures of ligands.

**Figure 2 molecules-24-01471-f002:**
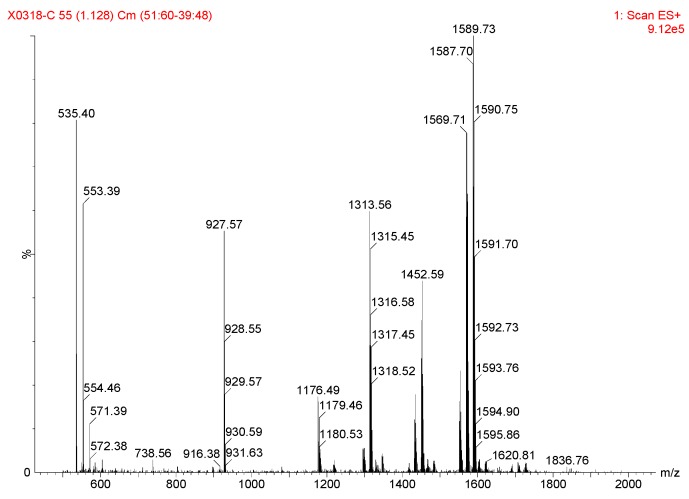
ESI-MS of Ni(OAc)_2_/**L2**/**1a** = 0.1/0.11/1.

**Figure 3 molecules-24-01471-f003:**
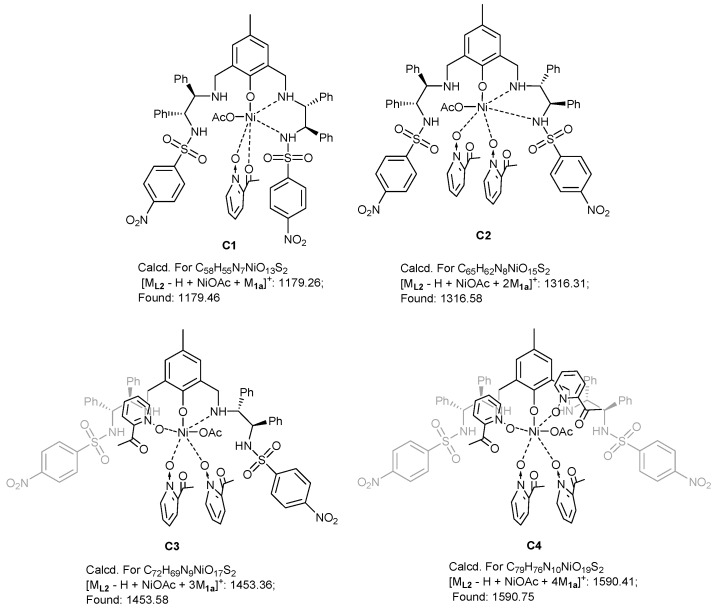
The speculated structures of Ni/**L2**/**1a** according to the ESI-MS analysis.

**Figure 4 molecules-24-01471-f004:**
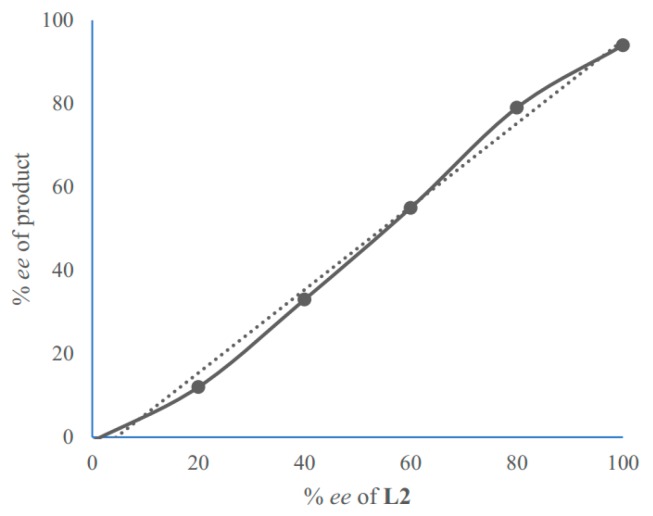
Linear relationship between *ee* of **L2** and *ee* of product **2**a.

**Figure 5 molecules-24-01471-f005:**
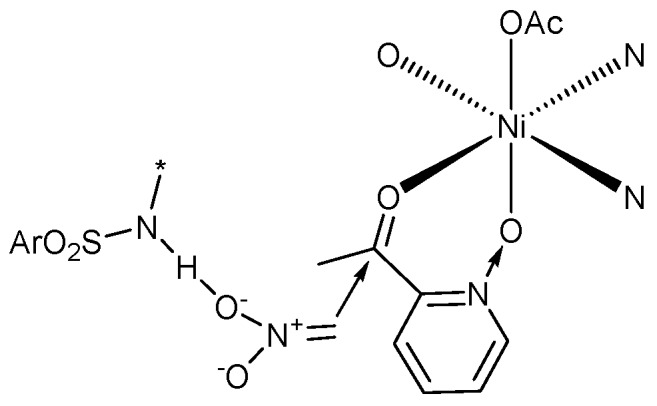
The proposed working model.

**Table 1 molecules-24-01471-t001:** Effect of the metal/ligand ratio and the ligand structure in the asymmetric Henry reaction ^a^.

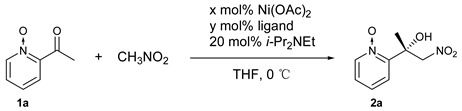					
**Entry**	**Ni(OAc)_2_ (x)**	**Ligand (y)**	**x/y**	**Yield (%) ^b^**	***ee* (%) ^c^**
1	20	**L1** (10)	2/1	94	76
2	15	**L1** (10)	1.5/1	98	81
3	10	**L1** (10)	1/1	81	81
4	10	**L1** (11)	1/1.1	86	85
5	10	**L1** (12)	1/1.2	79	83
6	10	**L1** (15)	1/1.5	72	84
7	10	**L1** (20)	1/2	73	83
8	10	**L2** (11)	1/1.1	99	91
9	10	**L3** (11)	1/1.1	90	83
10	10	**L4** (11)	1/1.1	91	76
11	10	**L5** (11)	1/1.1	89	89
12	10	**L6** (11)	1/1.1	50	4 ^d^
13	10	**L7** (11)	1/1.1	82	76
14	10	**L8** (11)	1/1.1	92	92
15	10	**L9** (11)	1/1.1	88	7
16	10	**L10** (11)	1/1.1	69	5 ^d^
17	10	**L11** (11)	1/1.1	98	15

^a^ Reactions were carried out with 2-acylpyridine *N*-oxides (0.2 mmol) with *i*-Pr_2_NEt (20 mol%) in a mixture of THF (0.8 mL) and CH_3_NO_2_ (0.2 mL) for 20 h. ^b^ Isolated yield. ^c^ Determined by chiral HPLC. ^d^ The absolute configuration of the major product was inverse compared with the others by the analysis of HPLC.

**Table 2 molecules-24-01471-t002:** Further optimization of the reaction ^a^.

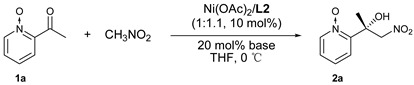			
**Entry**	**Base**	**Yield (%) ^b^**	***ee* (%) ^c^**
1	*i*Pr_2_NEt	99	91
2	Cyhex_2_NMe ^d^	99	94
3	Et_3_N	99	89
4	NMM ^e^	87	86
5	*i*Pr_2_NH	99	81
6	Bu_2_NH	99	88
7	Cyhex_2_NH ^f^	99	88
8 ^g^	Cyhex_2_NMe ^d^	99	83

^a^ Reactions were carried out with 2-acylpyridine *N*-oxides (0.2 mmol) with base (20 mol%) in a mixture of THF (0.8 mL) and CH_3_NO_2_ (0.2 mL) for 15–20 h. ^b^ Isolated yield. ^c^ Determined by chiral HPLC. ^d^
*N,N*-Dicyclohexylmethylamine. ^e^
*N*-methylmorpholine. ^f^ Dicyclohexylamine. ^g^ Different reaction operation: the order of addition of nitromethane and 2-acylpyridine *N*-oxide was reversed. In the standard operation, 2-acylpyridine *N*-oxide was added to the complex prepared in situ for 10 min before the addition of nitromethane. For the detailed standard operation, see the experimental section.

**Table 3 molecules-24-01471-t003:** Substrate scope for the asymmetric Henry reaction ^a^.

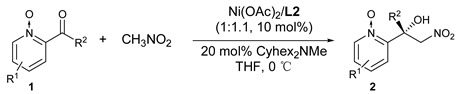						
**Entry**	**R^1^**	**R^2^**	**Product**	**Time (h)**	**Yield (%) ^b^**	***ee* (%) ^c^**
1	H	Me	**2a**	15	99	94
2	4-Me	Me	**2b**	15	99	99
3	5-Me	Me	**2c**	15	99	97
4	6-Me	Me	**2d**	24	99	17
5	4-Cl	Me	**2e**	17	99	92
6	5-Br	Me	**2f**	72	26	84
7	H	Et	**2g**	42	86	92
8	H	Bu	**2h**	42	67	69
9	H	4-ClC_6_H_4_	**2i**	72	48	79

^a^ Reactions were carried out with 2-acylpyridine *N*-oxides (0.2 mmol) with*N,N*-dicyclohexyl-methylamine (20 mol%) in a mixture of THF (0.8 mL) and CH_3_NO_2_ (0.2 mL). ^b^ Isolated yield. ^c^ Determined by chiral HPLC.

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
