# Peer review of "Asymmetric Henry Reaction of 2-Acylpyridine N-Oxides Catalyzed by a Ni-Aminophenol Sulfonamide Complex: An Unexpected Mononuclear Catalyst"

_molecules, 2019, doi:10.3390/molecules24081471_

Round 1
Reviewer 1 Report
The authors report on the asymmetric catalytic Henry reaction of 2-acylpyridine N-oxides with nitromethane. The reaction takes place in several instances with almost quantitative yields and also with ee's (up to 99% ee). The authors have recently reported on the same reaction (ref. 23) using a different catalyst, with inferior results regarding the enantioselectivity. Now, using complexes derived from Ni and multidentate chiral ligands of the aminophenolsulfonamide type, the reaction has been clearly improved. It is worth noting that with the exception of the work of Pedro et al. (ref. 40) there are no other precedents for this process. Although catalysts derived from this type of ligands are dimetallic complexes, the authors show that in this case the catalytic species have a 1:1 metal:ligand ratio, and this finding is rationalized by the high tendency of the substrates to complex the Ni atom through the basic N-oxide oxygen lone pairs.
These results are certainly interesting and significant, and deserve to be published in molecules.
Prior to publication, however, several English changes are required. Cf. among others Abstract: l.14 "has ben developed", not "had been developed"; l. 15 "mechanistic studies suggest" not "suggested". l.38 "the corresponding results are not satisfactory" (please develop), and experimental, l. 128 "The mixture continued to stir", change to "The mixture was further stirred", "the resulting solution purified by column chromatography" , the verb is missing, etc..
Other comments: Superindex g is missing in Table 2, entry 8.
Author Response
Point 1: Prior to publication, however, several English changes are required. Cf. among others Abstract: l.14 "has ben developed", not "had been developed"; l. 15 "mechanistic studies suggest" not "suggested". l.38 "the corresponding results are not satisfactory" (please develop), and experimental, l. 128 "The mixture continued to stir", change to "The mixture was further stirred", "the resulting solution purified by column chromatography" , the verb is missing, etc..
Response 1: The English mentioned by referee 1 has been changed. For detail, see the marked version of revised manuscript.
Point 2: Other comments: Superindex g is missing in Table 2, entry 8.
Response 2: The superindex “g” has been added in Table 2, entry 8.
Reviewer 2 Report
The paper entitled " Asymmetric Henry Reaction of 2-Acylpyridine N-Oxides Catalyzed by Ni-Aminophenol Sulfonamide Complex: An Unexpected Mononuclear Catalyst” by M. Liu et al. presents a short optimization study for the asymmetric Henry reactions with nitromethane and (substituted) 2-acyl-pyridine N-oxides as substrates. These investigations are a consequent extension of work published by the same group (ref.23, 41, 44-46) and others. Since catalysts and products are known (ref 40) the degree of novelty is moderate. At the other hand, this optimization study was conducted with caution, optimizing base, solvents and relative concentrations of reagents and some variation of substrate structure was done. The procedure is operationally simple giving in many cases excellent yields and good levels of enantioselectivity. From this point of view, the results might be of general interest and eventually will promote further investigation in this field.
Generally, the manuscript is clearly written with traceable conclusions so that the reader can follow the attempts to stepwise modify and improve reaction parameters. Merely, the validity of the assumption of a mononuclear Ni complex being the carrier of the process seems rather hypothetic at this stage as it is only based on HRMS evidence and trends in enantioselectivity. The references are sufficient.
After correcting some minor misprints (see below) I recommend publication of the manuscript in the present state.
page 2 line 52 change .... Entries.... to ....entries.....
page 3 line 76: a footnote “g” is mentioned but not in the table ?
page 4 line 98: m/z 1593 is not in agreement with Figure 2 and 3 (should be 1590?)
Could authors increase size of Figure 2?
page 5 Figure 3: in C3 and C4 “background substituents” are too pale to be clearly readable. Eventually thicker bonds could be used to mark foreground parts of the structure.
line 116: Models of instruments used should be given: NMR, polarimeter, HPLC.
page 6 line 126: Add “mmol” for 2-acylpyridine N-oxide.
line 128: no extractive workup?
For all compound names: Change 1-Methyl-2-Nitro-..... to 1-Methyl-2-nitro-
In several cases specific rotations of authors differ significantly from published values (ref. 40)! Any explanation?
page 7 line 191: should be ..... pyridine-containing ß-nitro tert-alcohols ......
Some wrong layout in references: 7, 28
Author Response
Point 1: page 2 line 52 change .... Entries.... to ....entries.....
Response 1: The misprints mentioned by referee 2 has been corrected. For detail, see the marked version of revised manuscript.
Point 2: page 3 line 76: a footnote “g” is mentioned but not in the table ?
Response 2: The superindex “g” has been added in Table 2, entry 8.
Point 3: page 4 line 98: m/z 1593 is not in agreement with Figure 2 and 3 (should be 1590?)
Response 3 Page 4, line 98: “1593” has been corrected as “1590”.
Point 4: Could authors increase size of Figure 2?
Response 4: The size of Figure 2 has been adjusted.
Point 5: page 5 Figure 3: in C3 and C4 “background substituents” are too pale to be clearly readable. Eventually thicker bonds could be used to mark foreground parts of the structure.
Response 5: The color of “background substituents” in C3 and C4 in Figure 3 has been adjusted according referee 2’s suggestion.
Point 6: line 116: Models of instruments used should be given: NMR, polarimeter, HPLC.
Response 6: Models of instruments used has been added.
Point 7: page 6 line 126: Add “mmol” for 2-acylpyridine N-oxide.
Response 7: “0.2 mmol” has been added for 2-acylpyridine N-oxide in line 127 page 6.
Point 8: line 128: no extractive workup?
Response 8: The reaction mixture was directly used in column chromatography without further treatment.
Point 9: For all compound names: Change 1-Methyl-2-Nitro-..... to 1-Methyl-2-nitro-
Response 9: The “1-Methyl-2-Nitro-…” in the compound names has been changed to “1-Methyl-2-nitro-…”.
Point 10: In several cases specific rotations of authors differ significantly from published values (ref. 40)! Any explanation?
Response 10: We recalculated the specific rotations (the picture of the original data is shown below) and we are sorry to found that there were some mistakes: (a) a typo with the concentration of 2b (the concentration of 2b should be c 0.4 instead of c 0.7); (b) a calculation mistake with the specific rotations of 2b (the specific rotations of 2b should be +156 instead of +164); (c) a typo with the specific rotations of 2d (the specific rotations of 2d should be +21 instead of +81).
At the time of preparing this manuscript, we found that the specific rotations of ours differ from published values (ref. 40). We think this is related to the ee of the compounds. It is speculated that the relationship between the ee and optical specific rotations of such compounds are not linear.
This Manuscript | Ref 40 | |||
ee (%) | Optical rotations | ee (%) | Optical rotations | |
2b | 99 | +156 (The original error calculation value is +164) | 84 | +41 |
2c | 97 | +181 | 81 | +60 |
Point 11: page 7 line 191: should be ..... pyridine-containing ß-nitro tert-alcohols ......
Response 11: “amino” has been changed to “nitro”.
Point 12: Some wrong layout in references: 7, 28
Response 12: We think the references 7 and 28 are reasonable and appropriate and we do not find any mistake in the two references.

Reviewer 3 Report
This manuscript by Zhou and co-workers reports the asymmetric Henry reaction of 2-acylpyridine N-oxide derivatives with Ni catalyst. Although asymmetric Henry reaction has been actively studied, 2-acylpyridine N-oxide is still a challenging substrate. The ligand used in this manuscript has been investigated as a ligand for di-nuclear complexes, while the authors interestingly found that mono-nuclear Ni complex was better for asymmetric Henry reaction of 2-asylpyridine N-oxide. In addition, the ee values of the products were generally good to high in this reaction system. Therefore, the referee supports publication of this manuscript in Molecules.
To improve this manuscript, the authors are recommended to revise the following points:
1) In all cases, more than 18 equivalents of MeNO2 was used. Please mention the results of the reactions with 1, 5, 10 equivalent(s) of MeNO2 in the main text.
2) In Figure 3: The authors showed some candidate complexes for MS spectra. However, the described complexes should be neutral species. Please confirm this point. In addition, the calibration of ESI-MS should be needed, taking into account of the MS-peaks and the ratio of isotopes.
3) If the authors have the data of the reaction with nitroethane, please add the result in this manuscript.
Author Response
Point 1: In all cases, more than 18 equivalents of MeNO2 was used. Please mention the results of the reactions with 1, 5, 10 equivalent(s) of MeNO2 in the main text.
Response 1: A related description has been added in line 81 page 3.
Point 2: In Figure 3: The authors showed some candidate complexes for MS spectra. However, the described complexes should be neutral species. Please confirm this point. In addition, the calibration of ESI-MS should be needed, taking into account of the MS-peaks and the ratio of isotopes.
Response 2: The complexes in Figure 3 are neutral species. And the discussion about the calibration of ESI-MS and the ratio of isotopes have been added in the Supporting Information (page S6-S9).
Point 3: If the authors have the data of the reaction with nitroethane, please add the result in this manuscript.
Response 3: The corresponding results of the reaction with nitroethane are not satisfactory.